# Anticancer Function and ROS-Mediated Multi-Targeting Anticancer Mechanisms of Copper (II) 2-hydroxy-1-naphthaldehyde Complexes

**DOI:** 10.3390/molecules24142544

**Published:** 2019-07-12

**Authors:** Muhammad Hamid Khan, Meiling Cai, Jungang Deng, Ping Yu, Hong Liang, Feng Yang

**Affiliations:** State Key Laboratory for the Chemistry and Molecular Engineering of Medicinal Resources, Guangxi Normal University, Guilin 541004, Guangxi, China

**Keywords:** Cu(II) complex, 2-hydroxy-1-naphthaldehyde, cytotoxicity, anticancer mechanism

## Abstract

Multi-targeting of oncoproteins by a single molecule represents an effectual, rational, and an alternative approach to target therapy. We carried out a systematic study to reveal the mechanisms of action of newly synthesized Cu^2+^ compounds of 2-naphthalenol and 1-(((2-pyridinylmethyl)imino)methyl)- (C1 and C2). The antiproliferative activity of the as-synthesized complexes in three human cancer cell lines indicates their potential as multi-targeted antitumor agents. Relatively, C1 and C2 showed better efficacy in vitro relative to Cisplatin and presented promising levels of toxicity against A-549 cells. On the whole, the Cu^2+^ complexes exhibited chemotherapeutic effects by generating reactive oxygen species (ROS) and arresting the cell cycle in the G_0_/G_1_ phase by competent regulation of cyclin and cyclin-dependent kinases. Fascinatingly, the Cu^2+^ complexes were shown to activate the apoptotic and autophagic pathways in A-549 cells. These complexes effectively induced endoplasmic reticulum stress-mediated apoptosis, inhibited topoisomerase-1, and damaged cancer DNA through a ROS-mediated mechanism. The synthesized Cu^2+^ complexes established ROS-mediated targeting of multiple cell signaling pathways as a fabulous route for the inhibition of cancer cell growth.

## 1. Introduction

In developing countries, cancer is the leading cause of death and is responsible for severe worldwide health and economic problems [1]. Platinum-based anticancer drugs, including Cisplatin and its analogs, are regarded as the most promising anticancer agents that have been used for the last few decades; however, targeted therapy, toxicity, and acquired resistance limit its clinical applications [2,3]. To reduce the toxicity and broaden the spectrum of activities of anticancer agents, multiple targeted anticancer agents could be promising anticancer candidates. On the other hand, transition metal complexes simultaneously act as multiple targeted agents [4,5,6].

Among transition metals, copper can be an alternative to platinum [1,7], as it plays a dynamic role in cancer chemotherapy. As a result, many copper therapeutics have been developed, some of which are currently being trialed as potential new anticancer agents [8]. Copper plays a vital role in many physiological activities of different tissues and organs [9,10], while also acting as a co-factor of several enzymes [11]. The primary anticancer mechanism of Cu(II) complexes is the generation of intracellular reactive oxygen species (ROS) by reduction of Cu(II) to Cu(I) [12,13,14]. Reactive oxygen species are highly reactive O_2_ metabolites, generated in the inner membrane of mitochondria and released into the intermembrane space and mitochondrial matrix [15]. Reactive oxygen species consist of hydroxyl radicals (HO•), superoxide radicals (O_2_•−), and hydrogen peroxide (H_2_O_2_) and act as critical regulators of, and secondary messengers in, various cell signaling pathways, including ROS-mediated autophagy and apoptosis [16,17,18].

Copper(II) complexes of 2-naphthalenol,1-(((2-pyridinylmethyl)imino)methyl)- have been designed and their structural characterizations identified [19,20] and investigated for their biological applications, which showed rational antibacterial activities [21]. Pyridine and its derivatives have a variety of biological applications. Metal agents with substituted pyridine exhibited better biological activities than the ligand without pyridine [22,23].

In this study, we focused on developing two Cu^2+^ 2-naphthalenol,1-(((2-pyridinylmethyl)imino)methyl)- with modifications of co-ligand pyridine (C1 and C2) (Figure 1B), and verified by mass-spectrometry, infrared spectroscopy, X-ray crystallography, and biological experiments. We investigated the multi-targeting mechanisms of Cu^2+^ complexes and show that they stimulate the production of ROS, induce mitogen-activated protein kinase (MAPK)-mediated apoptosis, mitochondrial-mediated apoptosis, and cell autophagy. In addition, these Cu^2+^ complexes potentially induce ROS-mediated endoplasmic reticulum stress-mediated apoptosis, inhibit topo-I activity, and damage DNA. 

## 2. Results

### 2.1. Development and Structure of Cu^2+^ Complexes

In this study, we designed and developed two Cu^2+^ compounds containing 2-naphthalenol,1-(((2-pyridinylmethyl)imino)methyl)-, (C1), and modified with co-ligand pyridine (C2). As shown in Figure 1B, these compounds were prepared with copper salt reaction including 2-hydroxy-1-naphthaldehyde,(2-(aminomethyl) pyridine) and pyridine in methanol using the stirred and refluxed method at 65 °C for 3 h. The crystals were collected using the direct method. The crystal structures were identified by X-ray crystallography (Figure 1A).

C1 was crystallized in the space group P21/n making a monoclinic structure. The molecular structure of the arrangement of the Cu(II) metal center is listed in Table 1. The Cu(II) metal center was pentacoordinate with two nitrogen atoms and an oxygen atom of the ligand and two terminal chlorine atoms. The distances of the Cu–N/O bonds ranged from 1.913–2.02 Å (Appendix A). The coordination of the polyhedron surrounding the Cu atom at the center could be displayed by a square pyramid, with the metal displaced from the N1/N2/O basal plane, ranging from 88–175.9° (Appendix A).

C2 was crystallized in the space group P21/n making a monoclinic structure. The coordination geometry in C2 was approximately the same as in C1. Cu in C2 was pentacoordinate with two nitrogen atoms and one oxygen atom of the ligand, one nitrogen atom of the pyridine substituent, and one terminal bromide atom. The distances of the Cu–N/O/Br bonds ranged from 1.917–2.85 Å. The Cu (II) metal center assumed a square–pyramidal coordination geometry (Appendix A). Due to the narrow-angle of the O1–Cu1–Br unit in C2, a slight strain existed in the square–planar coordination plane around the Cu(II) center and was displaced from N1/N2/O basal plane, ranging from 82.48–177.06° (Appendix A).

### 2.2. Integrality of Cu^2+^ Complexes in Solution

To confirm the Cu^2+^ complexes’ resistance to degradation in phosphate buffer over time, we analyzed the samples by UV-vis spectra and investigated in defined solutions at room temperature. In brief, the Cu^2+^ complexes were dissolved in PBS at room temperature, and then their respective UV-visible spectra were tested at different times (0–48 h). As shown in Appendix A, there was no obvious red or blue shift in the absorption peaks of each group, and no new absorption peaks appeared. These results indicate that the structure of the Zn complexes did not change in the solution and was not degradable in the solution for 48 h.

### 2.3. Anticancer Properties of C1 and C2

#### 2.3.1. Comparative Cytotoxicity of C1 and C2 In Vitro

The three cell lines A-549, MGC-803, and T-24, and the normal cell line HL-7702 were used to study the cytotoxicity of the two Cu^2+^ complexes in vitro. The 3-(4,5-dimethylthiazol-2-yl)-2,5-diphenyltetrazolium bromide (MTT) assay was used to determine the IC_50_ values of the Cu^2+^ compounds (Table 2). Notably, C1 and C2 exhibited lower IC_50_ values against A-549 cells of 1.06 ± 0.01 µM and 0.7 ± 0. 01 3 µM, respectively, than that of cisplatin (17.36 ± 0.25 μM) [24], but showed lower cytotoxicity against HL-7702 cells (11.02 ± 0.15 and 10.05 ± 0.12, respectively). These results suggested that the Cu^2+^ complexes were found to be more selective towards tumor cells versus normal cells, and potentially inhibited the tumor cells more than cisplatin. Comparative to C1, C2 showed higher cytotoxicity against A-549 cells in vitro. Cu^2+^, and more likely the pyridine, enhanced the formation of ROS in cancer cells, which are accountable for the higher cytotoxicity of the designed compound.

To investigate the intracellular Cu and Pt contents in the cancer cells, inductively coupled plasma (ICP-MS) was carried out. The A-549 cells were treated with C1, C2, and cisplatin for 24 h. The nuclear and cytoplasmic fraction was extracted and separated using the NE-PER reagent kit. The total intracellular contents of Cu and Pt in C1, C2, and Cisplatin samples were 18 nmol, 21 nmol, and 15 nmol, which constituted 33%, 38%, and 27%, respectively (Figure 2B). Similarly, the cytoplasmic contents of Cu and Pt in C1, C2, and cisplatin samples were measured at 10.98 nmol, 12.81 nmol, and 9.1 nmol, while in the nucleus, they were observed at 7.02 nmol, 8.19 nmol, and 5.9 nmol, respectively. Comparatively, the Cu contents found in the C2 samples were higher than that of Cu in C1 and Pt in cisplatin cell samples. The co-ligand pyridine bound to the C2 could influence the concentration of complex matter being absorbed to the cells and could enhance the cytotoxicity of the Cu^2+^ complexes.

#### 2.3.2. Comparative Influence of C1 and C2 on Spheroid Growth

Compared with conventional two-dimensional (2D) cell culture, three-dimensional (3D) cell culture is considered more accurately approximate to the body environment and could enable more accurate evaluation of drug efficacy [25]. After treatment of A-549 cells with C1 and C2, the inhibition of A-549 tumor spheroids was investigated for seven days. We found that untreated spheroids increased in volume, while treated spheroids exhibited a more compact shape. The tumor cells on the surface of treated tumor spheroids were relatively disorganized (Figure 3). The results suggest that the C1 and C2 kept the spheroids’ structure compact and inhibited tumor progression in A-549 cells.

### 2.4. ROS-Mediated Multi-Targeted Anticancer Mechanisms of Cu^2+^ Compounds

#### 2.4.1. Analysis of the Stimulation of ROS in A-549 Cells

Excess ROS induce oxidative alteration of cellular macromolecules, inhibits protein function, and promotes apoptosis by intracellular and extracellular signals [26,27]. Intracellular ROS is primarily generated in mitochondria, where 1–2% of the total consumed O_2_ by mitochondria is diverted to ROS, which consist of O_2_•−, HO•, and H_2_O_2_ radicals [28,29]. In the presence of metal ions, high reactive HO• is generated, causing significant damage to cellular protein, lipids, and DNA [30].

The 2′,7′-dichlorofluorescein diacetate (DCF-DA) assay kit was used to determine the ROS level in A-549 cells. The treated and untreated A-549 cells were incubated in DCF-DA for 30 min. After incubation, the fluorescence intensity was investigated by flow cytometry showing that C1 and C2 complexes generated ROS in A-549 cells (Figure 2A). The results suggested that the treated cells had stronger DCF fluorescence intensity and had a deviated peak.

#### 2.4.2. Regulation of the Cell Cycle

The flow cytometry analysis data revealed that the percentage of untreated A-549 cells in the G_0_/G_1_ phase was 52.82%. When treated with C1 (1.4 µM), it was increased to 57.09%. However, when it was treated with C2 (1.4 µM), it was raised further to 70.30% (Figure 4A). These results suggest that Cu^2+^ complexes accumulated in the A-549 cells during the G_0_/G_1_ phase of the cell cycle. It has been demonstrated that ROS production in mitochondria is accompanied by cytochrome-c release and activation of apoptotic pathways, the LC3II-mediated autophagic pathway, and topo-I mediated DNA damage [31,32]. On the other hand, cyclin E is the regulatory cyclin for CDK2 and is considered a vital regulator of G_1_ phase progression. Cyclin E is overexpressed in cancer, suggesting that cyclin E/CDK2 deregulation contributes to tumorigenesis [33,34]. The Western blot results suggested that the Cu^2+^ complexes inhibited the expressions of cyclin E and CDK2, which support the above flow cytometry analysis that can effectively generate ROS and regulate the expression of proteins related to the G_0_/G_1_ phase in A-549 cells (Figure 4B,C).

#### 2.4.3. Study of Mitochondrial apoptosis in A-549 induced by Cu^2+^ Complexes

B cell lymphoma (Bcl-2) family proteins play a vital role in mitochondrial-mediated apoptosis. During the stressed condition, Bcl-2 family proteins promote mitochondrial membrane permeabilization and induce mitochondrial-mediated apoptosis [35,36]. ROS remarkably activate the proapoptotic protein via the activation of Bad and Bax and by the inhibition of Bcl-2 protein expression, resulting in a loss of Δψm and a release of cytochrome-c. The cytochrome-c further activates casp-9 and casp-3, which leads to cell apoptosis [37,38,39].

Annexin-V/PI double-staining assay revealed the occurrence of apoptosis in A-549 cells while being treated with C1 and C2 complexes. The flow cytometry analysis data showed that the rate of apoptosis in A-549 cells treated with C1 (1.4 µM) was 28.4%, and this was increased to 33.3% while treated with C2 (1.4 µM) (Figure 5A). To determine the mitochondrial Δψm, the JC-1 mitochondrial potential assay kit was used. The Δψm decreased to 27.9% and 39.1% respectively, suggesting the depolarization of the mitochondria, inducing apoptosis in cancer cells and providing firm evidence that C1 and C2 work through the apoptosis pathway (Figure 6).

Furthermore, Western blot data quantify the Bax, Bcl-2, Bcl-xl, and cytochrome-c in C1- and C2-treated A-549 cells (Figure 5B,C). Results showed that the expression of Bax and cytochrome-c were upregulated, whereas Bcl-2 and Bcl-xl were downregulated. In addition, caspase-9 and caspase-3 were observed to be cleaved in A-549 cells, indicating the activation of mitochondrial apoptosis in cancer cells (Figure 5B,D). Such data suggest that C1 and C2 could effectively stimulate ROS-mediated mitochondrial apoptosis by regulating the expression of Bcl-2 family proteins.

#### 2.4.4. ROS-Mediated Endoplasmic Reticulum Stress Pathway

The endoplasmic reticulum (ER) plays a critical role in assembling, folding, and transporting cellular proteins. The function of the ER is Ca^2+^ dependent; imbalance of Ca^2+^ levels in the ER results in ionic or redox states, which lead to the accumulation of unfolded proteins. Cells regulate ER stress by activating the unified signal transduction pathway called the unfolded protein response. Persistent in ER stress leads to cell apoptosis [40,41,42]. In response to ER stress, the protein kinase RNA-like ER kinase (PERK) inactivates the inhibitory factor eIF2α through phosphorylation and reduces protein synthesis [43,44]. On the other hand, in stress conditions, the CCAAT-enhancer-binding protein homologous protein (CHOP) is activated, which further activates mitochondrial proapoptotic proteins [45]. Figure 7A confirms that, compared to untreated A-549 cells, the cells treated with C1 and C2 showed more deviated peaks, indicating the imbalance of Ca^2+^ concentration in A-549 cells. Moreover, the Western blot data reveal the activation of PERK, eIF2α, and CHOP in A-549 cells (Figure 7B,C).

The A-549 cells were analyzed, after treatment with C1 and C2, using 2ʹ,7ʹ-dichlorodihydrofluorescein diacetate (H2DCFDA), which was oxidized by cellular ROS to 2ʹ,7ʹ-dichlorofluorescein (DCF), while the ER stress was analyzed using ER-tracker red to bind to Sulfonylurea receptors in the ER. After incubating the A-549 cells with C1 and C2, cleared colonization was observed in the cultured cells, which were further increased with an increasing incubation time up to 18 h. The ROS were found to be more diffused and less colonized in the cultured A 549 cells, which may indicate the increasing permeabilization of the ER membrane (Figure 8).

#### 2.4.5. Mitogen-Activated Protein Kinase (MAPK) Pathway Study

Apoptosis signaling-regulated protein kinase (ASK-1) is a MAP-kinase family protein, also called mitogen-activated protein kinase (MAP3K5). In response to ROS, ASK-1 activates the MAPK pathway [46]. MAPK is a chain of proteins that communicate with cell surface receptors and DNA in the nucleus, which control cell division. MAPK is also called extracellular signal-regulated kinase (ERK). An extracellular mitogen is phosphorylated, which helps in the activation of MAPKs. MAPKs further activate transcriptional factors like cyclins and cyclin-dependent kinases [47]. P38MAPKs, ERKs, and JNKs are the isoforms of MAPK family proteins [48,49,50]. In response to cellular stress, including ROS, these proteins are activated by phosphorylation, and their cell functions, including apoptosis, are activated [51]. Western blot data suggest that C1 and C2 remarkably induced the accumulation of JNKs, p38MAPKs, and ERKs in A-549 cells, indicating the ROS-mediated MAPK signaling apoptosis in A-549 cells (Figure 9A,B).

#### 2.4.6. Study of Autophagy in A-549 Cells

ROS exert intracellular stress and induce cells’ autophagy. During autophagy, the light chain-3 (LC3) protein is converted into cytosolic form (LC3-I), which is bound to phosphatidylethanolamine to form a complex called LC3-II. At the same time, LC3-I enables the docking of specific cargos protein (p62), which is then degraded, following an increase in autophagic reflex and formation of autolysosome. The intra-autolysosomal components are decomposed, their building blocks are released from the vesicles by the action of permeases, and lead to cell autophagy [52,53,54,55]. Similarly, beclin-1 is an autophagic-associated protein, which regulates cell autophagy and apoptosis. Beclin-1 interacts with Bcl family proteins and regulates the cell cycle [56]. In brief, the A-549 cells were treated with C1 and C2 for 24 h and analyzed by Western blot for autophagic-associated proteins. The results indicated that C1 and C2 remarkably upregulated the LC3-II and beclin-1, and downregulated the p62 proteins, suggesting the induction of autophagy in A-549 cells (Figure 2C,D).

#### 2.4.7. Topoisomerase I Inhibition

ROS can induce topoisomerase-mediated cell apoptosis, which is an important mechanism contributing to cancer cell death [57]. DNA-topoisomerases represents an essential family of DNA-processing enzymes, and some topoisomerase inhibitors are used clinically for the treatment of numerous human cancers [58]. A gel electrophoresis method was used to determine the topo-I cleavage in vitro. It was observed that the Cu^2+^ compounds significantly inhibited the cleavage of the topo-I enzyme, resulting in the inhibition of DNA replication and lead to cells apoptosis (Figure 10C).

#### 2.4.8. ROS-Mediated DNA Binding and Cleavage Study

Although high levels of cellular ROS can significantly damage DNA and could affect the physiological activities of cells, cells have an elaborate system of DNA repair [59]. The goal of chemotherapeutics to simultaneously cause DNA damage and inhibit DNA repair could be the most promising anticancer approach [60]. The interaction of the Cu^2+^ compounds with DNA was analyzed by UV-Vis absorption spectroscopy. The absorption spectra of C1 and C2 in the presence of different concentrations of calf thymus DNA (ct-DNA) were examined and recorded by instant absorption bands of around 370–470 nm. After successive additions of ct-DNA, the absorption peaks of C1 and C2 increased and exhibited approximately 10% hypochromisms. These spectroscopic changes suggested that there is an interaction between Cu2^+^ complexes (C1 and C2) and ct-DNA (Figure 11A,B).

Furthermore, ethidium bromide (EB) is the intercalating compound of DNA. We investigated the competitive binding capacity of Cu^2+^ compounds with a DNA–EB system. We observed that C1 and C2 could compete with the DNA–EB system and could effectively bind to ctDNA by replacing EB molecules (Figure 11C,D).

The ability of the Cu^2+^ compounds to split the supercoiled pBR322 plasmid DNA was analyzed by agarose gel electrophoresis. Higher concentrations of C1 and C2 relaxed the supercoiled DNA into circular form (Figure 10A). Importantly, to assess the ROS-mediated DNA damage, we exposed the DNA to C1 and C2 in the presence of H_2_O_2_ (3 mM) and analyzed it using gel electrophoresis (Figure 10B). The DNA exposed to H_2_O_2_ (lane 1) and CuCl_2_ + H_2_O_2_ (lane 2) showed that it was unaffected, mean that O_2_•− and HO• were not involved in DNA cleavage. When the DNA was treated with C1, the reaction of the DNA cleavage was partially inhibited (lane 3 and lane 4). While the DNA was treated with C2, the cleavage was significantly inhibited (lane 5 and lane 6), implying that H_2_O_2_ was the reactive species in the cleavage process and DNA might be decomposed by the oxidation of hydroxyl radicals (HO•) produced by the Cu^2+^ compounds, particularly by C2.

## 3. Discussion

Targeted therapy aims at delivering drugs to specific genes or proteins that are particular to cancer cells and do not affect healthy cells [61,62]. However, poor selectivity, acquired resistance, and adverse side effects intensely limit their applications [63,64]. To this end, the multi-targeted anticancer approach could be the ultimate method of targeted delivery, which could minimize these limitations by inhibiting multiple cell signaling pathways, and which could encourage the discovery of novel multi-mechanism metal anticancer agents [65,66,67].

Discovery of novel multi-targeted anticancer agents from 2-hydroxy-1-naphthaldehyde is an area receiving increasing amounts of consideration [68,69]. Our experiments assessing cellular uptake showed that complexes modified with a co-ligand pyridine were readily absorbed and intracellularly distributed in A-549 cells, which may indicate the mechanisms underlying the increase in activity of Cu(II) complexes in our studies.

Our results demonstrated that Cu^2+^ compounds inhibit the growth of A-549 cells by targeting multiple signaling pathways. These complexes exert a chemotherapeutic effect by the formation of ROS and induced the apoptotic and autophagic pathways, including mitochondrial-mediated apoptosis, ER stress-mediated apoptosis, mitogen-activated protein kinase pathway (MAPK) pathway, regulation of autophagic markers (LC3-II, p62, and beclin-1), and inhibition of cleavage of topo-I and DNA damage. Consequently, the Cu(II) complexes may hold promise as a novel multi-targeted metal anticancer drug candidate.

## 4. Experiments

### 4.1. Material and Methods

All chemicals used were in highly pure forms. 2-hydroxy-1-naphthaldehyde and (2-aminomethyl) pyridine) were provided by Energy Chemical Company (Shanghai, China). All cell lines (A-549, MGC-803, T-24, and HL-7702) were provided by American Type Culture Collection and the German Collection of Microorganisms and Cell Cultures. Reactive oxygen species kit, JC-I kit, and NE-PER were purchased from Beyotime (Jiangsu, China). The antibodies used in Western blotting were provided by Abcam (Cambridge, UK).

### 4.2. Development of Cu^2+^ Complexes

#### 4.2.1. Synthesis of Ligand (L)

The ligand was synthesized according to the previously reported protocol [70]. 2-hydroxy-1-naphthaldehyde and (2-(aminomethyl) pyridine) (0.5 mmol) were dissolved in a 20 mL aqueous methanol solution and stirred for 3 h at 65 °C, yielding a dark brown solution, yield: 79.6%. Anal. Calcd (%) for C17H14N2O: C, 77.81; H, 5.37; N, 10.67; O, 6.09. Found: C, 77.84; H, 5.38; N, 10.68; O, 6.10. IR (main peaks cm^−1^) 3435, 3045, 2923, 2003, 1637, 1619, 1544, 1471, 1374, 1234, 1117, 993, 942, 890, 770, 583. *m/z* (ESI): calcd for C17H14N2O, 263.318 (H)^+^ (Appendix A).

#### 4.2.2. Synthesis of C1

L and CuCl2 (0.5 mmol) were stirred for 1 h at 65 °C to give a clear solution and then filtered. The filtrate was kept in the air for a week, forming blue-black crystals. The crystals were directly isolated, washed three times with distilled water, and dried in a vacuum desiccator containing anhydrous CaCl2. Yield: 83%. Anal. Calc for C17H13Cl2CuN2O (395.75): C, 51.59; H, 3.31 and N, 8.68. Found: C, 51.51; H, 3.24 and N, 8.60. IR (Main Peak cm^−1^): 3461.33, 3090.05, 1719, 1601.92, 1344.26, 761.53, 702.21. MS *m/z*: 396.75 (M+H)^+^ (Appendix A).

#### 4.2.3. Synthesis of C2

L (0.5 mmol), CuBr2 (0.5 mmol), and pyridine (0.5 mmol) were stirred for 1 h at 65 °C and then filtered. The filtrate was kept in the air for one week, forming blue-black crystals. The crystals were directly isolated, washed with distilled water, and dried in a vacuum desiccator containing anhydrous CaCl_2_. Yield: 83%. Anal. Calc for C_22_H_18_BrCuN_3_O (483.84): C, 54.61; H, 3.75 and N, 8.68. Found: C, 54.55; H, 3.68 and N, 8.62. IR (Main Peak cm^−1^): 3443.25, 3010.24, 1626.12, 1440.24, 1342.73, 1025.10, 954.43, 707.48. MS *m/z*: 484.0 (M+H)^+^ (Appendix A).

#### 4.2.4. X-Ray Crystallography of C1 and C2

Single crystals of C1 and C2 were analyzed with a Bruker APEX-II CCD diffractometer. During the data collection, the crystals were kept at 296.15 K. The structures were solved using Olex2 [71] (Table 1). The data in the CIF format were input at the Cambridge crystallographic data center under the CCDC deposition number 1038424 (C1) and 1938378 (C2). The selected bond lengths (Å) and bond angles (°) of the crystal structure are listed in Appendix A.

### 4.3. Anticancer Activity of the C1 and C2 Complexes

#### 4.3.1. Cell Culture

Adenocarcinoma human alveolar basal epithelial cells (A-549), human gastric cancer cell line MGC-803, human urinary bladder cancer (T24), and immortalized human hepatocyte (HL-7702) cells were purchased from the American Type Culture Collection and the German Collection of Microorganisms and Cell Cultures. Cells were incubated at 37 °C under the humidified atmosphere containing 5% CO_2_ and cultured in DMEM supplemented with 10% fetal bovine serum (FBS) and 1% penicillin/streptomycin.

#### 4.3.2. Cytotoxicity Assay (MTT)

To determine the IC_50_ value of Cu(II) complexes, 3-(4,5-dimethylthiazol-2-yl)-2,5-diphenyltetrazolium bromide (MTT) assay was performed. In brief, 180 µL of cell suspensions at 5 × 10^4^ cells/mL were cultured in 96 well plates and incubated for 24 h at 37 °C in 5% CO_2_ environment. Test compounds were added with the specified concentration into each well except the control and incubated for 48 h at 37 °C and 5% CO_2_ condition. The assay was initiated by adding 10 µL of MTT to each well, and the plates were incubated for another 4 h. The medium was removed, and 100 μL/well DMSO was added to dissolve the resulting blue formazan crystals. Enzyme labeling instruments with 570/630 nm dual wavelength measurements analyzed the absorbance spectra of the wells. The cytotoxicity was assessed based on the percentage of cell survival compared with the control.

#### 4.3.3. Tumor Spheroid Growth Inhibition

The three-dimensional spheroid is helpful to study the physiological characteristics of cells and can differentiate the tumor cells from normal cells. In brief, 96 well microtiter plates were used to perform this procedure. 1.5 % (*wt/v*) agarose DMEM solution was heated at 80 °C and sterilized. 50 µL of agarose solution was added to each microtiter well and allowed to solidify. 1 × 10^3^ cells/well of A-549 cell were cultured on the upper lips of cell culture plates and incubated in 37 °C, 5% CO_2_ atmosphere. After 72 h, each of the A-549 cells’ spheroid was transferred into each agarose-coated well and allowed to settle. Each A-549 spheroid, except the control, was treated with C1 and C2 and analyzed every 24 h for spheroid growth (Figure 3).

### 4.4. Determination of Possible Anticancer Mechanisms of Cu^+2^ Compounds

#### 4.4.1. Cell Cycle Distribution Analysis

Propidium iodide (PI) staining method was used to analyze the cell cycle distribution in A-549 cells. Briefly, A-549 cell cultures were treated with C1 (1.4 µM) and C2 (1.4 µM) and incubated for 24 h. Cells were collected by centrifugation, washed with cold PBS, and fixed with 70% ethanol overnight at −20 °C. The cells were then incubated with DNase-free RNase (100 µg/mL) for 30 min, followed by PI staining in the dark for 30 min. The cells were analyzed for cell cycle phases by flow cytometry (FACScan, Bection Dickinson, San Jose, CA).

#### 4.4.2. Apoptosis by Flow Cytometry

Annexin-V/Propidium iodide (Annexin V-FITC Apoptosis Detection Kit) double-staining procedure was used to assess the occurrence of apoptosis in A-549 cells. A-549 cells were cultured and treated with C1 (1.4 µM) and C2 (1.4 µM) for 24 h. The cells were collected by centrifugation and resuspended in 200 µL Annexin-binding buffer, followed by the addition of 5 µL Annexin-V and 4 µL of PI. The cells were then incubated at 37 °C in the dark for 15 min. After adding 300 µL of Annexin-binding buffer, the cells were analyzed by flow cytometry (FACScan, Bection Dickinson, San Jose, CA).

#### 4.4.3. Mitochondrial Membrane Potential Assay

A JC-1 (Beyotime, Haimen, China) probe was used to analyze the mitochondrial depolarization in A-549 cells. A-549 cells were treated with C1 (1.4 µM) and C2 (1.4 µM) for 12 h and collected by centrifugation. The cells were resuspended in 0.5 mL of JC-1 (10 µg/mL) and incubated for 30 min at 37 °C in the dark. The cells were then analyzed by flow cytometry (FACScan, Bection Dickinson, San Jose, CA).

#### 4.4.4. Intracellular ROS Assay

2′,7′-dichlorodihydrofluorescein diacetate (H2DCF-DA) was used to determine the intracellular ROS in cancer cells. Briefly, 1 × 10^5^ cells/well were cultured and treated with C1 (1.4 µM) and C (1.4 µM) for 24 h at 37 °C. The cells were collected by centrifugation, followed by incubation with 500 µL of a stock solution of H2DCF-DA for 30 min. The cells were analyzed by flow cytometry with the fluorescence intensity wavelength at 488 nm and with the emission wavelength of 525 nm.

#### 4.4.5. Visualization of ROS in the Endoplasmic Reticulum

A-549 cells were cultured in six-well plates at 2 × 10^6^ cells/well on poly-L-lysine-coated coverslips in heat-deactivated complete RPMI. The cells were then treated with C1 and C2 for 6 h and washed with pre-warmed PBS, followed by incubation with 25 µM pre-warmed 2′,7′-dichloro-dihydro-fluorescein diacetate (H2DCF-DA) for 20 min at 37 °C. 1µM pre-warmed ER-tracker red was added to the cells and incubated for 15 min. The cells were then washed with PBS, mounted onto slides, and analyzed using confocal microscopy. Images were acquired using a 20× objective lens and processed using the Zeiss FLUOVIEW Viewer software.

#### 4.4.6. Detection of Intracellular Ca^2+^ Ions

To determine the intracellular Ca^2+^, fluo-3 AM was used. In brief, A-549 cells were cultured in six-well plates and incubated at 37 °C for 24 h, followed by incubation with C1 and C2 for a further 24 h. The cells were then incubated with fluo-3 AM for Ca^2+^ staining and analyzed using flow cytometry.

#### 4.4.7. Topoisomerase I Inhibition Assay

Human DNA topoisomerase-I (Topo-I) inhibitory activity was determined by measuring the relaxation of supercoiled plasmid DNA pBR322 using gel electrophoresis method. Each sample contained 10× DNA Topo-I buffer, 0.1 % BSA, 0.25 μg plasmid DNA pBR322, 1 Unit Topo-I, and Cu^2+^ compounds. The samples were incubated at 37 °C for 30 min, followed by adding 2 μL of 5× loading buffer (0.25 % bromophenol blue, 4.5% SDS, and 45% glycerol). 1% agarose containing 0.02 % (*v/v*) gold view at 8 V/cm was run for 1 h with 1× TAE as the running buffer. “GenoSens Gel Analysis-image capture” was used to capture the image of topo-I activities (Figure 10C).

#### 4.4.8. DNA Binding and Cleavage Study

The DNA binding and cleavage were analyzed by electronic absorption spectra ranging from the 200–700 nm wavelength. Each test sample contained Cu(II) complex, calf thymus DNA (ct-DNA), and equilibrated in pH 7.2 (5 mM Tris HCl/ 50 mM NaCl). Emission spectra ranging from 530–750 nm was recorded after each successive addition of ct-DNA (Figure 11A,B). Besides, the relative binding capacity of Cu^2+^ with ct-DNA-EB was analyzed by fluorescence spectrometry. In brief, ethidium bromide (EB) (8 µM) and ctDNA (10 µM) was dissolved in pH 7.2 Tri-HCl buffer, and after each successive addition of 5 mM C1 and C2, the wavelength of the fluorescence spectrometry was recorded (Figure 11C,D).

Agarose gel electrophoresis method was used to determine the potency of Cu^2+^ compounds cleavage of DNA. Each reaction mixture, in different concentrations in separate tubes, contained pBR322 DNA (0.5 µL), 2 µL of each of C1 and C2, tris-HCl buffer, and 2 µL of a 6× loading buffer. The reaction mixture was loaded in an agarose gel using 1× TAE as running buffer (Figure 10A). On the other hand, to determine the ROS-mediated DNA damage, a gel electrophoresis method was used. The effect of Cu^2+^ on pBR322 plasmid DNA was analyzed in the presence of H_2_O_2_. Each testing sample contained plasmid DNA + H_2_O_2_ (lane 1), plasmid DNA + H_2_O_2_ + CuCl_2_ (lane 2), plasmid DNA + H_2_O_2_ + C1 (Lanes 3 and 4), plasmid DNA + H_2_O_2_ + C2 (Lanes 5 and 6) (Figure 10B) “GenoSens Gel Analysis-image capture” was used to capture the images of each test sample.

#### 4.4.9. Western Blot Analysis

The A-549 cells were cultured and treated with C1 and C2 for 24 h. After incubation, the cells were collected by centrifugation. The cells were washed with cold PBS and lysed with radioimmunoprecipitation assay (RIPA) buffer supplemented with an inhibitor of proteases and phosphatase sodium orthovanadate. The bicinchoninic acid (BCA) was used to determine the protein concentration in the cell’s lysate. Total cellular proteins were separated using 10% SDS-polyacrylamide gel electrophoresis. The proteins were transferred onto polyvinylidene difluoride (PVDF) membrane and blocked with 5% non-fat milk in TBST buffer (20 mM Tris, pH 8.0, 0.05% Tween 20, and 150 mM NaCl) for two hours. The PVD membrane was incubated overnight with anti-mouse or anti-rabbit primary antibodies. The membrane was washed with TBST and incubated with secondary antibodies for two hours at room temperature. Amersham ECL plus was used to detect immunoreactivity of the protein with antibodies. Tanon™ 5200CE Chemi-Image System was used to detect the final images of the samples.

#### 4.4.10. NE-PER Assay

The A-549 cells were treated with C1, C2, and cisplatin. The cells were harvested, centrifuged, and washed with PBS. Cytoplasmic extraction reagent-I (CER-I), cytoplasmic extraction reagent-II (CER-II), and nuclear extraction reagent (NER) were used to detect the cytoplasmic and nuclear contents in 200:11:100 ratios. CER-I was added to the microcentrifuge tubes, while tube vortexes were checked for 15 s and placed on ice for 10 min. CER-II was added to the cell pellets followed by another for 15 s vortexing. Tubes were centrifuged and the supernatant transferred to the new micro-centrifuge tubes. This content had cytoplasmic extractions and was stored at −80 °C.

In the next step, NER was added to the cell pellets, followed by 15 s vortexing and placed on ice for 10 min. After repeating the vortex four times, the tubes were then centrifuged and the supernatants transferred to new micro-centrifuge tubes. These nuclear extraction-bearing samples were stored at −80 °C until analyzed by ICP-MS for copper and platinum contents in the nucleus and cytoplasm of A-549 cells.

## 5. Conclusions

The newly synthesized Cu(II) compounds 2-hydroxyl-1-naphthaldehyde and (2-(aminomethyl) pyridine) generated ROS and showed multi-targeted anticancer activities. We increased the cytotoxicity of these compounds by making complexes with a co-ligand pyridine. These complexes absorbed and accumulated in cancerous cells and killed A-549 cells through multi-targeted mechanisms. Our results demonstrate that the 1-(((2-pyridinylmethyl)imino)methyl)- complexes may be promising candidates for the treatment of adenocarcinomic human alveolar basal epithelial cell cancer.

## Figures and Tables

**Figure 1 molecules-24-02544-f001:**
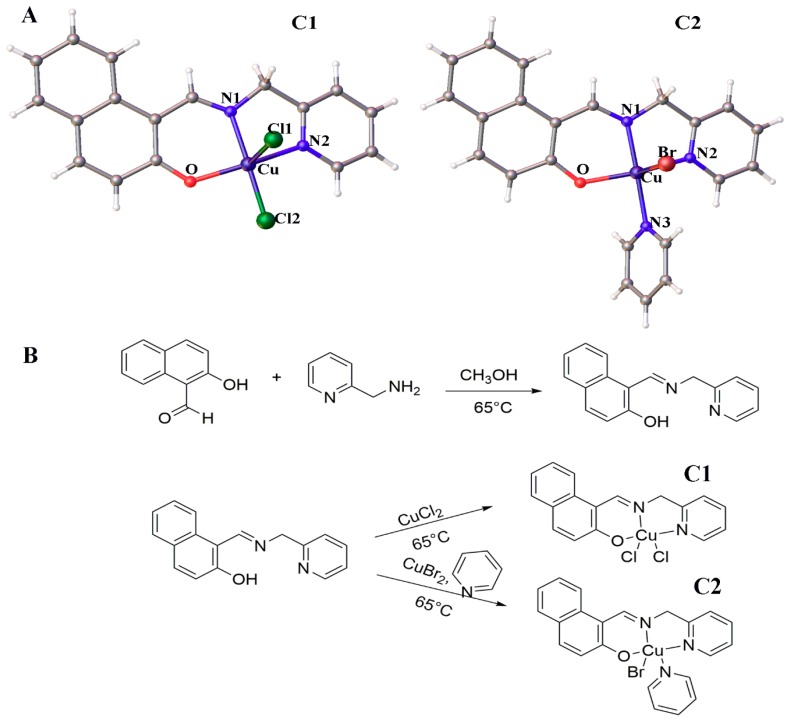
(**A**) Chemical structures of Cu^2+^ compounds. (**B**) Synthetic routes of Cu^2+^ compounds (C1 and C2).

**Figure 2 molecules-24-02544-f002:**
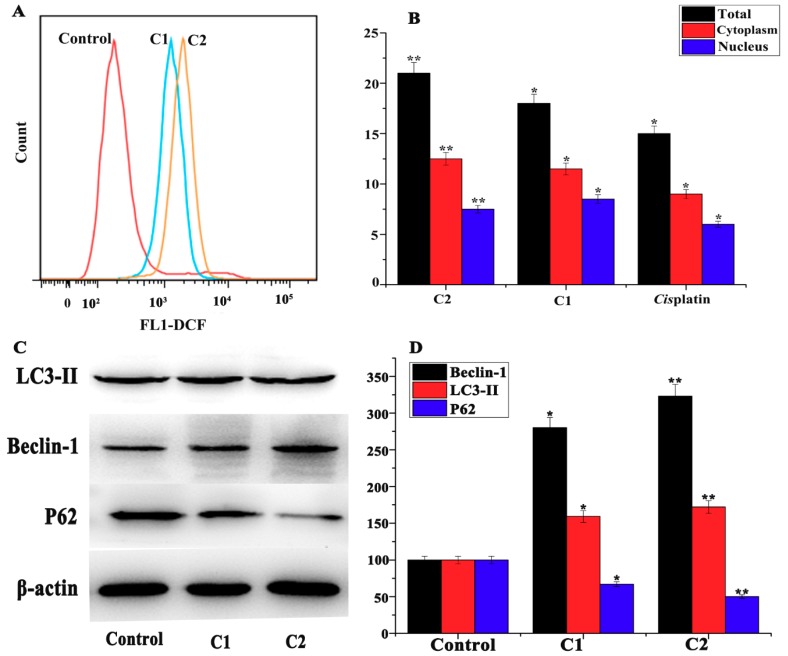
(**A**) Effect of ROS levels of A-549 cells treated with C1 (1.4 μM) and C2 (1.4 μM) for 24 h compared with untreated cells, quantification of the flow cytometric results. (**B**) Cu and Pt contents (%) in A-549 cells after treatment with C1 (1.4 μM) and C2 (1.4 μM) and analyzed by ICP-MS. (**C**) Western blot analysis of the expression level of LC3-II, Beclin-1, and P62 in A-549 cells treated with C1 (1.4 µM) and C2 (1.4 µM) relative to control. (**D**) The percentage expression level of the LC3-II, Beclin-1, and P62. Mean ± SD, *n* = 3, * *p* < 0.05, ** *p* < 0.01.

**Figure 3 molecules-24-02544-f003:**
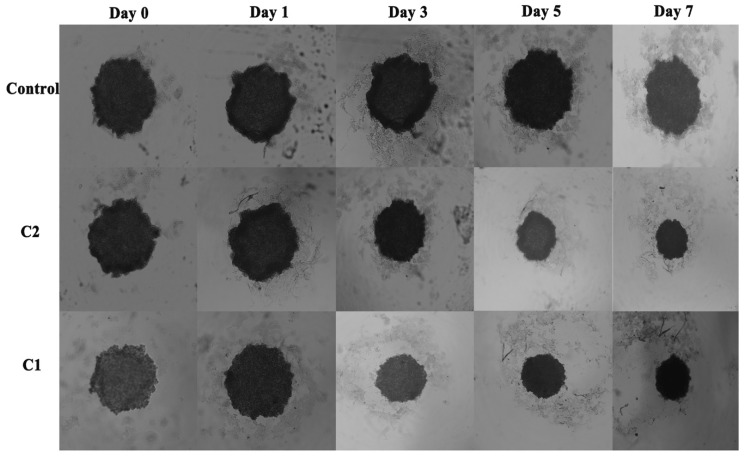
Tumor spheroid analysis of A-549 cells after being treated with C1 and C2 with specified concentrations for 7 days.

**Figure 4 molecules-24-02544-f004:**
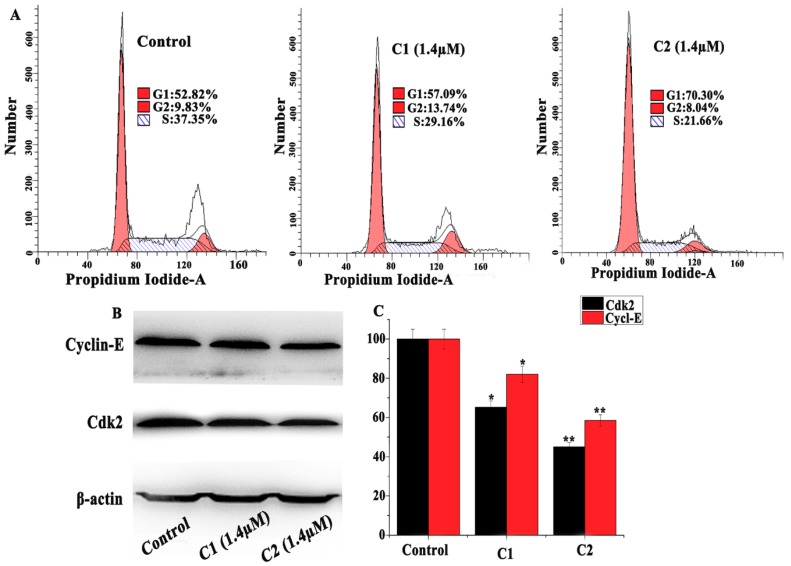
(**A**) Effect of the cell cycle of A-549 cells treated with C1 (1.4 µM) and C2 (1.4 µM) compared with untreated cells. (**B**) Western blot analysis of CDK2 and cyclin E in A-549 cells treated with C1 (1.4 μM) and C2 (1.4 μM). (**C**) Percentage expression levels of CDK2 and cyclin E. The percentage values are those relative to the control. Results are the mean ± SD, *n* = 3, * *p* < 0.05, ** *p* < 0.01.

**Figure 5 molecules-24-02544-f005:**
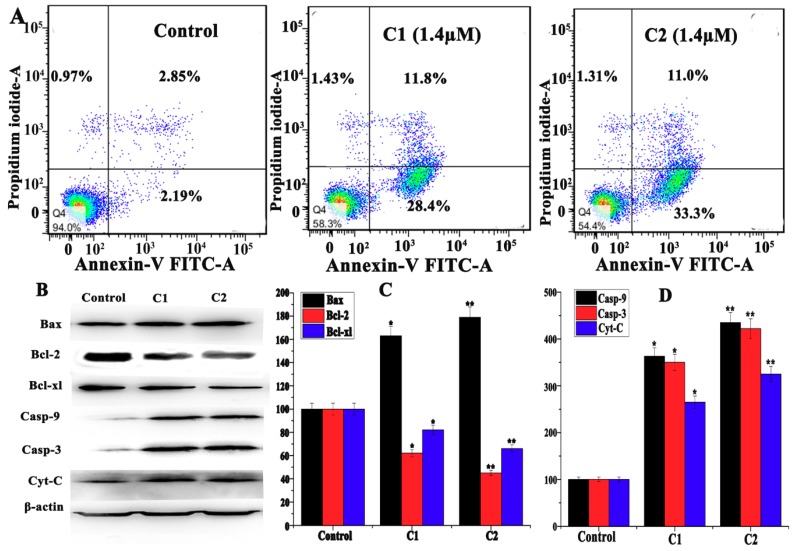
(**A**) Effect of cell apoptosis of A-549 cells treated with C1 (1.4 µM) and C2 (1.4 µM) comparing with untreated cells. (**B**) Western blot analysis of Bax, Bcl-2, Bcl-xl, caspase 3 (cleaved), Caspase 9 (cleaved), and Cytochrome-c in A-549 cells treated with C1 (1.4 µM) and C2 (1.4 µM). (**C**) The percentage expression level of Casp-3 (cleaved), Casp-9 (cleaved), and cytochrome-c relative to control. (**D**) The percentage expression level of Bax, Bcl-2, and Bcl-xl relative to its control value. Results are the mean ± SD, *n* = 3, * *p* < 0.05, ** *p* < 0.01.

**Figure 6 molecules-24-02544-f006:**
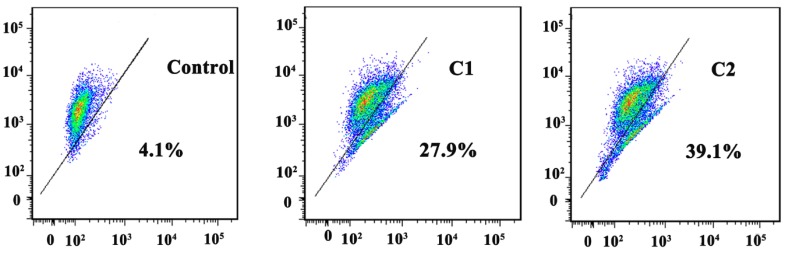
Assay of the A-549 cells’ mitochondrial membrane potential with JC-1 fluorescence probe staining compared with untreated cells.

**Figure 7 molecules-24-02544-f007:**
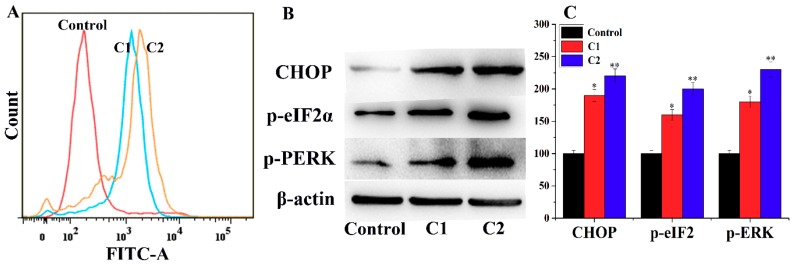
(**A**) A-549 cells treated with C1 and C2 and analysis of Ca^2+^ concentration. (**B**) Western blot analysis of PERK, eIF2α, and CHOP in A-549 cells. (**C**) Graphical presentation of the expression levels of PERK, eIF2α, and CHOP. Mean ± SD, (*n* = 3), * *p* < 0.05, ***p* < 0.01.

**Figure 8 molecules-24-02544-f008:**
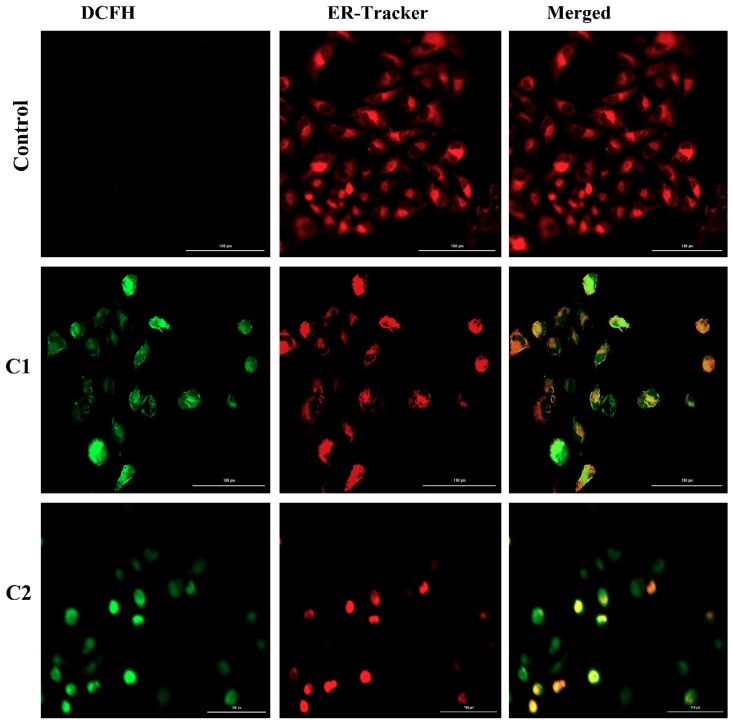
Generation of ROS by using 2′,7′-dichlorofluorescein (DCF) and analysis of ER-tracker red by binding to sulfonylurea receptors in the ER.

**Figure 9 molecules-24-02544-f009:**
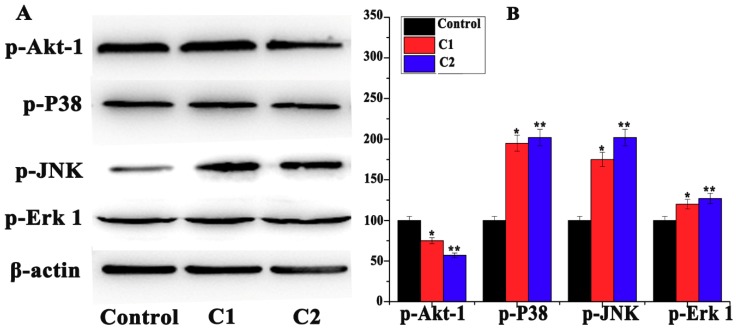
(**A**) Western blot analysis of Akt1/2, P38, JNKs, Erk-1/2 in A-549 cells treated with C1 (1.4 µM) and C2 (1.4 µM). (**B**) Percentage expression levels of Akt1/2, P38, JNKs, Erk-1/2 relative to their control value. Mean ± SD, *n* = 3, * *p* < 0.05, ** *p* < 0.01.

**Figure 10 molecules-24-02544-f010:**
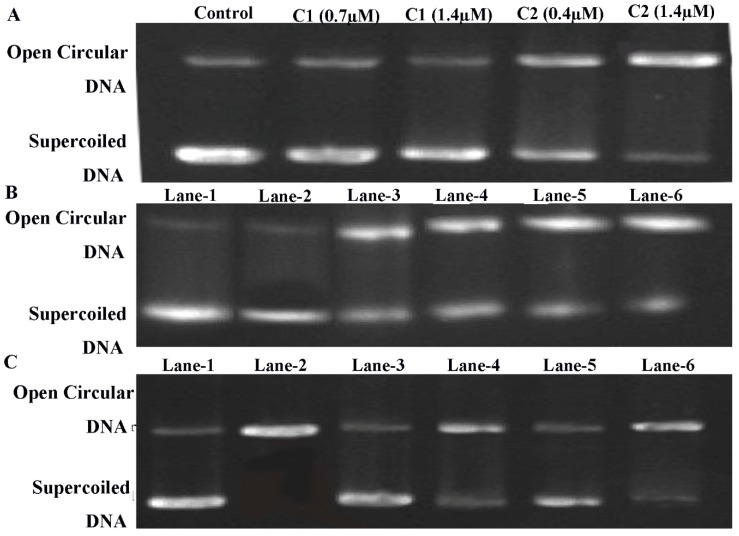
(**A**) Cleavage of pBR322 plasmid DNA by C1 and C2, using agarose gel electrophoresis. (**B**) Cleavage of pBR322 plasmid DNA by C1 and C2 in the presence of H_2_O_2_ using agarose gel electrophoresis; lane 1: Cleavage of pBR322 plasmid DNA by C1 and C2, using agarose gel electrophoresis. (**B**) Cleavage of pBR322 plasmid DNA by C1 and C2 in the presence of H_2_O_2_ using agarose gel electrophoresis; lane 1: Plasmid DNA + H_2_O_2_; lane 2: Plasmid DNA + H_2_O_2_ + CuCl_2_; lane 3: Plasmid DNA + H_2_O_2_ + C1 (0.7 µM); lane 4: Plasmid DNA + H_2_O_2_ + C1 (1.4 µM); lane 5: Plasmid DNA + H_2_O_2_ + C2 (0.7 µM); lane 6: Plasmid DNA + H_2_O_2_ + C2 (1.4 µM). **C**. Inhibition of topoisomerase-I activity; lane 1: Plasmid DNA; lane 2: Plasmid DNA + 1U topoisomerase I; lane 3: Plasmid DNA + 1U topoisomerase-I + C1 (0.7 µM); lane 4: Plasmid + 1U topoisomerase-I + C (1.4 µM); lane 5: Plasmid DNA + 1U topoisomerase-I + C2 (0.7 µM); lane 6: Plasmid DNA + topoisomerase-I + C2 (1.4 µM).

**Figure 11 molecules-24-02544-f011:**
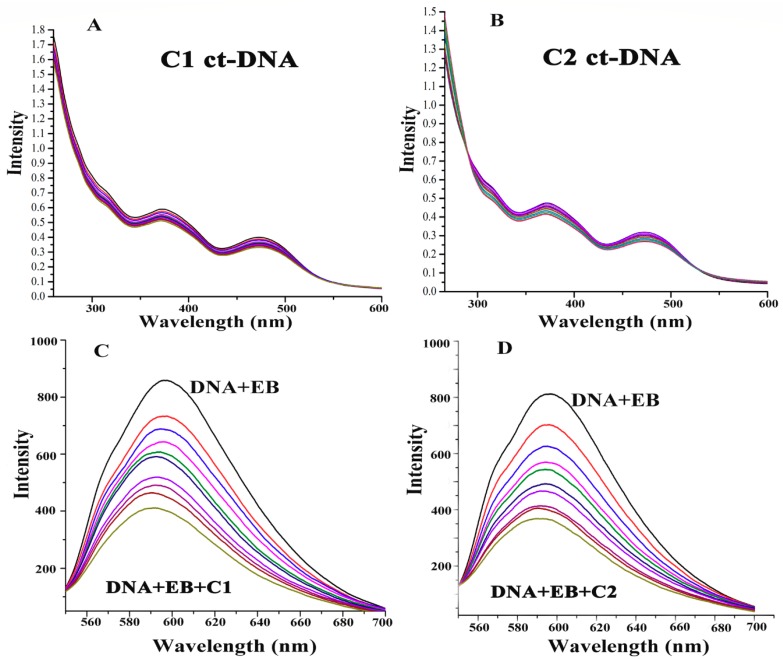
(**A**) UV-Vis titration of 5 µM C1 with ct-DNA in 10 mM Tris-HCl (7.4). (**B**) UV-Vis titration of 5 µM C2 with ct-DNA in 10 mM Tris-HCl (7.4). (**C**) Effect of C1 on the emission intensity of EB (8 µM) bound to ctDNA (10 µM) in pH 7.2 Tri-HCl buffer. (**D**) Effect of C2 on the emission intensity of EB (8 µM) bound to ctDNA (10 µM) in pH 7.2 Tri-HCl buffer.

**Table 1 molecules-24-02544-t001:** Data collection statistics and crystallographic analysis of Cu^2+^ compounds (C1 and C2).

Identification Code	C1	C2
Empirical formula	C_17_H_13_Cl_2_CuN_2_O	C_22_H_18_BrCuN_3_O
Formula weight	395.73	483.86
Temperature/K	296.15	296.15
Crystal system	monoclinic	monoclinic
Space group	P21/n	P21/n
a/Å	8.6992(9)	8.6992(9)
b/Å	13.6084(14)	13.6084(14)
c/Å	16.4250(18)	16.4250(18)
α/°	90	90
β/°	90.070(2)	90.070(2)
γ/°	90	90
Volume/Å3	1944.4(4)	1944.4(4)
Z	4	4
ρcalcg/cm^3^	1.352	1.6527
μ(Mo–Kα) mm^–1^	1.401	3.196
F(000)	800	972.6
Data/restraints/parameters	4211/0/208	4211/0/253
Goodness-of-fit on F2	4.817	1.027
Final R indexes (I>=2σ (I))	R1 = 0.2332,	R1 = 0.0376,
	wR2 = 0.5701	wR2 = 0.0920
Largest difference peak/hole /e Å3	7.60/−2.60	0.91/−0.46

**Table 2 molecules-24-02544-t002:** IC_50_ (µM) values of C1, C2, and Cisplatin toward cancer cells and normal cells for 48 h.

Compounds IC_50_ ± SD (µM)				
	A-549	MGC-803	T24	HL-7702
L	>50	>50	>50	>50
C1	1.06 ± 0.01	1.98 ± 0.14	2.21 ± 0.21	11.02 ± 0.15
C2	0.7 ± 0.01	1.17 ± 0.09	2.09 ± 0.26	10.05 ± 0.12
Cisplatin	21.74 ± 1.52	17.87 ± 0.35	22.21 ± 0.54	10.21 ± 0.24

IC_50_ values are presented as the mean ± SD from three separate experiments.

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
