# Peer review of "Anticancer Function and ROS-Mediated Multi-Targeting Anticancer Mechanisms of Copper (II) 2-hydroxy-1-naphthaldehyde Complexes"

_molecules, 2019, doi:10.3390/molecules24142544_

Round 1

Reviewer 1 Report

The manuscript by Muhammad Hamid Khan et al. reports the synthesis of two new copper(II) complexes with tridentate Schiff-base ligand, their crystal structures and detailed biological investigation related to their cytotoxicity. The manuscript is well written and could be of interest to biological inorganic chemistry audience.

However, several points need to be taken into account:

1. The ligand was reported before in the literature, so the authors should give a reference to its preparation procedure.

2. Several complexes of the same ligand are reported in the literature, including copper(II) complexes (doi 10.1002/zaac.200400270, 10.1107/S1600536804016265, 10.1016/j.ica.2013.10.035), in the introduction part the authors should give a brief overview of the compounds already reported and especially pay attention to copper(II) chloride complex reported in 10.1016/j.ica.2013.10.035, since in a very similar conditions to the ones used by the authors binuclear chloride-bridged complex was formed.

3. CCDC deposion numbers for the crystal structures of the complexes should be given in the manuscript.

4. ROS reactivity with copper(II) complexes was evaluated in papers 10.1007/s00775-006-0101-1 and 10.1039/b900869a, that the authors could consider citing in their manuscript.

Author Response

dear reviewer, please enclose the the attached files. 

thank you. 

Reviewer 2 Report

This work describes the synthesis and biological evaluation of two copper complexes with a tridentate Schiff base ligand. Extensive in vitro biological studies were performed, which is the strong part of this work.

The ligand has not been properly isolated and characterized.This presents a problem since this compound was used in the biological experiments (tested for its antitumor properties in cell lines). Therefore it is suggested to add these data. Also, it would be preferred to adopt its chemical name instead of  2-hydroxy-1-naphthaldehyde, [2- (aminomethyl) pyridine], which only describes the reagents it was made from.

In table 2, what is L1 and what L2? Please add this information.

Also, do the authors have any information on the stability if these two complexes C1 and C2 in the solvents used for the biological assays that were performed? If yes, please describe. For example the labile ligands may exchange in solution and therefore there two complexes C1 and C2 become the same species.

Paragraph 2.2.1, could be revised to clarify that the copper complexes were found to be more selective towards tumor cells vs normal cells in comparison to cisplatin.

The sentence in lines 121-122 was not well understood, please revise.

In paragraph 2.3.2 please describe the role of cyclin E and Cdk2 in cell cycle to explain the results shown in Figure 4B and 4C.

In paragraph 2.3.3, lines 155-160 appear in small fonts. Was that intentional? Please revise.

Figure 6 should be separated from text and its legend formatted as the others.

In Figure 11 B and C was the experiment  conducted in increasing concentration of C1 and C2? If so, please add this information in the legend, or wherever appropriate.

In the experimental section, define what aqueous methanolic solution is (e.g.1:1 MeOH:water?)

Author Response

(The authors gave the same response as above.)

Round 2

Reviewer 2 Report

The authors revised the manuscript according to the recommendations.